

# Black Carbon, Organic Carbon, and Co-Pollutants Emissions and Energy Efficiency from Artisanal Brick Production in Mexico

Miguel Zavala[1], Luisa T. Molina[1], Pablo Maiz[2], Israel Monsivais[2], Judith C. Chow[3], John G. Watson[3], Jose Luis Munguia[4], Beatriz Cardenas[5], Edward C. Fortner[6], Scott C. Herndon[6], Joseph R. Roscioli[6], Charles E. Kolb[6], Walter B. Knighton[7]

[1]Molina Center for Energy and the Environment, La Jolla, CA, 92037, USA
[2]GAMATEK, Monterrey, Nuevo Leon, Mexico
[3]Desert Research Institute, Las Vegas, NV, 89119, USA
[4]Universidad Autónoma Metropolitana, Mexico City, Mexico
[5]Secretaria del Medio Ambiente, Mexico City, Mexico
[6]Aerodyne Research, Inc., Billerica, MA, 01821, USA
[7]Department of Chemistry and Biochemistry, Montana State University, MT, 59717, USA.

*Correspondence to*: Luisa Molina (ltmolina@mce2.org; ltmolina@mit.edu)

**Abstract.** In many parts of the developing world and economies in transition, small-scale traditional brick kilns are a notorious source of urban air pollution. Many are both energy inefficient and burn highly polluting fuels that emit significant levels of black carbon (BC), organic carbon (OC) and other atmospheric pollutants into local communities, resulting in severe health and environmental impacts. However, only a very limited number of studies are available on the emission characteristics of brick kilns; thus there is a need to characterize their gaseous and particulate matter (PM) emission factors to better assess their overall contribution to emissions inventories and to quantify their ecological, human health, and climate impacts. In this study, the fuel-, energy-, and brick-based emissions factors and time-based emission ratios of BC, OC, inorganic PM components, CO, $SO_2$, $CH_4$, $NO_x$, and selected volatile organic compounds (VOCs) from two traditional artisanal kilns and one MK2 kiln in Mexico were quantified using the tracer ratio sampling technique. Simultaneous measurements of PM components, CO and $CO_2$ were also obtained using a filter-based sampling probe technique. Additional measurements included the internal temperature of the brick kilns, mechanical resistance of bricks produced, and characteristics of fuels employed. The results show that both techniques capture similar temporal profiles of the brick kiln emissions and produce comparable emission factors, indicating that the tracer ratio technique can be an alternative option to the filter-based sampling probe technique in understanding the temporal profile of the chemical composition of brick kilns emissions. A more integrated inter-comparison of the brick kilns' performances was obtained by simultaneously assessing emissions factors, energy efficiency, fuel consumption, and the quality of the bricks produced. Overall, a well-designed and operated MK2 kiln produced lower $PM_{2.5}$, BC, OC emission factors and higher energy efficiency than the traditional artisanal brick kilns. Average fuel-based BC emission factors ranged from 0.15 – 0.58 g/kg-fuel whereas BC/OC mass ratios ranged from 0.9 - 5.2, depending on the kiln type. The results from this study contribute to the limited number of databases available on the emission characteristics of the informal brick production sector.



## 1 Introduction

Artisanal clay brick production using small-scale traditional kilns is a highly polluting activity occurring in developing countries and economies in transition to manufacture building materials. Moreover, traditional brick production is a serious local health hazard to the residents of the poor neighborhoods that typically host brickyards, as well as to brick makers
themselves. Impacts of toxic emissions on brick producers' respiratory health and the environment have been documented in a number of studies (e.g., Zuskin et al., 1998; Co et al., 2009; Martínez-Salinas et al., 2010; Kaushik et al., 2012). Although production zones are clustered at the periphery of -or even within- urban areas, laborers and their families often lack access to adequate public services including clean water, basic sanitation facilities, health services, transport, and education infrastructure. Brick producers often sell the bricks to intermediaries and the economic revenue for producers can be marginal.
These conditions contribute to the perpetuation of severe environmental and social injustice problems.

The most current estimates suggest that about 1.5 trillion clay bricks are produced annually, with 90% of the global production generated by Asian countries, and with only a small fraction (less than 10%) of global brick production using modern mechanized technology (CIATEC, 2015). However, being predominantly an informal industrial sector, there are substantial
uncertainties in the number, types, fuels, and characteristics of kilns used for this activity. The lack of reliable activity data and emission factors makes it difficult to quantify the overall contribution of brick production to local and regional emissions inventories and to assess the ecological, human health and climate impacts.

Efforts in Mexico to reduce the impacts of bricks production include the promotion of technologically improved kilns and
survey-based field studies to improve the activity data for this sector (Cardenas et al., 2012). The few data available indicate that fuels and the characteristics of raw materials vary based on their cost and availability. The estimated number of brick kilns in Mexico is about 17,000, of which 75% are "traditional-fixed" type with permanent walls that delimit the space of accommodation of the bricks to be cooked; 22% are "traditional-campaign" kilns in which the raw bricks give shape to the kiln, and only < 3% are mechanically industrialized or of new design (CIATEC, 2015). One of the new designs is a double
dome version of the original Marquez Kiln (MK) developed by R. O. Marquez (2002) called MK2 which involves covering the kiln with a dome and channeling the output flow through a second loaded kiln for its additional filtration of the effluents (Bruce et al., 2007). However, there is a need for an integrated assessment of the emissions and energy performance of traditional and new kiln designs as well as the identification of the economic, social and technical barriers to adopt new technologies by brick producers (Schmidt, 2013).

The general steps of brick production include clay preparation, molding, drying, and firing. The firing process itself is divided into burning, smoldering, and cooling stages. Nevertheless, the whole process is artisanal rather than standardized, learned by experience, and locally adjusted depending on the soil characteristics, kiln design, and available fuels. In Mexico, biomass is





the predominant fuel used in the production of bricks, although it is often combined with other hazardous and highly polluting materials including waste oils, textiles, tires and plastics (CIATEC, 2015). This results in low efficiency combustion and high levels of gaseous and particulate matter (PM) pollutants that are difficult to quantify in an emissions inventory.

Brick kiln emissions are suspected to be a major source of black carbon (BC) and other PM components at the local scale in developing countries. However, there are no reliable estimates of global emissions from brick kilns. Based on a very limited number of measurements and expert judgment, Bond et al. (2013) estimated that industrial coal combustion provided about 9% of global BC emissions in 2000, although that figure includes brick production as well as small boilers, process heating for lime kilns, and coke production for the steel industry. In Mexico, the 2008 National Emissions Inventory (2008-MNEI)
suggests emissions of 2.9, 0.5, and 19.7 Gg of $PM_{2.5}$, $NO_x$, and volatile organic compounds (VOCs), respectively, from brick kilns (SEMARNAT, 2012). Nevertheless, these estimates were obtained using emission factors from the AP-42 US-EPA database that may not apply to kiln technologies and operating conditions in Mexico. There is a need to reduce the uncertainties associated with the estimation of emissions from brick production.

A limited number of studies exist on the emission characteristics of brick kilns. Le and Oanh (2010) measured the emission rates of CO, $SO_2$ and PM in two kilns in Vietnam. Christian et al. (2010) measured the emission factors of multiple gases and PM composition, including BC and organic carbon (OC), from three traditional brick kilns in Mexico. Maiz et al. (2010) determined emission factors for several types of dioxins, furans and other persistent organic pollutants (POPs) from two types of artisanal brick kilns. Umlauf et al. (2017) determined various POPs in soil, bottom ash and products from brickmaking sites
in Kenya, Mexico and South Africa. Fifteen kilns in India and two in Vietnam representing five types of kiln designs were sampled for their CO, $CO_2$, $SO_2$, (Rajarathnam et al., 2014) and their $PM_{2.5}$ and elemental carbon (EC) emission factors and optical properties (Weyant et al., 2014). Stockwell et al. (2016) measured a "zig-zag" kiln and a batch-type clamp kiln burning coal as fuel in Nepal to obtain emission factors for a large suite of gases and PM composition. Overall, the results from these studies indicate that emission factors are highly variable and depend on fuel type, feeding patterns, fraction of internal and
external fuel, and kiln designs. Despite the widespread use of brick kilns in Latin American countries there have been very limited studies on the emission impacts of kiln designs and fuels employed.

Due to the intensity of the emission fluxes, the high temperatures involved, and the varied geometry of the kilns, there are considerable technical challenges associated with the measurement of emission factors from brick kilns. Recently, based on a
review of the available studies, the Climate and Clean Air Coalition (CCAC) Brick Production Initiative has developed guidelines for the measurement of brick kilns emissions and energy performance (Weyant et al., 2016). The guidelines include procedures for the isokinetic probe sampling of effluents in kiln stacks when they are available, and the use of an array probe in the open plume above the kiln to apply the carbon mass balance method (Thomson et al., 2016).



As part of the pilot field measurement campaign to characterize the emissions from key sources of Short-Lived Climate Forcers in Mexico (SLCF-2013 Mexico), we measured the emissions factors for BC, OC, the inorganic PM components, CO, $SO_2$, $NO_x$, $CH_4$, and selected VOCs from a traditional-fixed kiln, a traditional-campaign kiln, and a MK2 kiln in Mexico using a tracer ratio method sampling technique, allowing the examination of the emission plume's evolution as it transits downwind

from the source. The tracer ratio method (Lamb et al., 1995) has been used to measure emissions from other similar types of industrial and area sources. To our knowledge, this technique had never been applied for measuring emissions from brick production. Simultaneous measurements of PM components, CO and $CO_2$ were obtained using the sampling probe technique, thus allowing a unique comparison between the two different techniques. Additional measurements included the internal brick kilns temperature, energy efficiency, mechanical resistance of bricks produced, and chemical composition of fuels employed.

The emissions were measured both during the firing and subsequent smoldering stages, providing insight into the effects of different kiln designs and fuels on gaseous and particulate phase emissions from brick kilns.

## 2 Methodology

### 2.1 Brick kilns sampled

Table 1 lists the characteristics of the brick kilns sampled and Fig. 1 shows the three kilns during measurements. A description

of their operation processes is presented in the Supplemental Material document. The MK2 kiln and the traditional-campaign kiln were measured in El Refugio, a community of brick producers located in the periphery of Leon, Guanajuato. The traditional-fixed kiln was measured in a separate community of brick producers in Abasolo, Guanajuato. Measurements took place during the dry season on March 12-16, 2013. Close collaboration with the local authorities and the brick producers' associations allowed us to establish an agreement that other kilns would not be fired during the measurement period to

minimize the influence from nearby sources. The selected kilns were operated by experienced brick producers under real-world operating conditions, with fuels types and practices they commonly use.

A random sample of 60 bricks were identified, measured, and weighed before the firing took place for each kiln. At the end of the firing, these same bricks were again measured, weighed and sent to a laboratory to test their mechanical resistance and

water absorption content following the corresponding NMX-CC-404-ONNCCE-2012 Mexican standard (ONNCCE, 2012). Samples of fuels and raw materials were collected before the firing to determine carbon content and heating value of combustion for the fuels. The determination was carried out with an Elemental Analyzer PE-2400 Series1 and a microbalance. An acetanilide standard was used to calibrate the equipment and obtain the sample's carbon content. Heating value of combustion was determined using ASTM standards (ASTM 1995) with a Parr-1108 calorimetric pump operating with excess

of oxygen to assure complete combustion of the sample. The results of these analyses are presented in Tables SM1-SM3 in the Supplemental Material document. Thermocouples were installed at the lower, middle, and upper levels of the kiln to determine the temperature profiles inside the kiln during its operation. These three levels were defined as follow: 0.4 m above the





combustion chamber for the lower level, 0.3 m below the last layer of bricks for the upper level, and half the distance between the lower and upper levels for the middle level.

## 2.2 Sampling techniques

Two sampling techniques were used to obtain the emission factors of pollutants generated from the brick kilns. In the sampling-probe technique, a temporary scaffold was built on the side of the kiln for equipment and technicians, and a probe was installed on top of the kiln and connected to a sensor sampling train containing real-time sensors and filters for PM collection. This sampling technique is possible due to the relatively low velocities of the exhaust so that an isokinetic flow train is not required (Weyant et al., 2016). During the few seconds right after exiting the kiln and before they are well mixed downwind, the

emission plumes on top of the kiln can vary substantially in intensity and composition. This implies that the location of the sampling probe on top of the kiln is of key importance to the representativeness of the filter measurement. To account for this effect, the sampling probe was mounted on a rotating crane that was continuously spinning slowly on top of the kiln (see Fig. 1).

An inertial mass separator with a cut-point of 2.5 µm was used to obtain the $PM_{2.5}$ fraction of PM collected on 47-mm diameter quartz filters. The $PM_{2.5}$ filters were replaced approximately once an hour depending on the pressure drop on the sampler. After the samplings, the filters were thermally stabilized and sent to the laboratory for gravimetric and EC and OC composition analysis using thermal/optical transmittance (TOT) and reflectance (TOR) analysis (Chow et al., 2004) using the IMPROVE_A protocol (Chow et al., 2007). Although the EC measured by the thermal/optical methods is not technically considered as BC

(Petzold et al., 2013), in this paper we refer to EC by TOR as a surrogate of BC as the light-absorbing carbon in the measured PM. Since the collection filters were heavily loaded and had homogenous deposits, analysis of anions (chlorides, nitrates, sulfates) and cations (ammonium and water- soluble sodium and potassium) analyses by ion chromatography were performed. Laboratory analyses showed that field blank concentrations were low in relation to those in source samples, averaging < 5 % for OC and < 0.1% for BC.

Exhaust flow in the sampling train was measured using a piston flowmeter and directed to a Continuous Emissions Monitoring System (CEMS) with a Fourier-Transform Infrared Spectrometer (FTIR) to measure $CO_2$ and CO and to a flame ionization detector (FID) analyser to measure total gaseous organic compounds (TOG). Instrument specifications and sampling calibration protocols are described in Tables SM4-SM5 in the Supplemental Material document. The gaseous carbon

concentration in standard conditions are used in the carbon mass balance method together with the measured carbon content of the fuels (see Tables SM2-SM3) to obtain fuel-based emission factors ($EF_{fuel}$, g/kg-fuel) of a pollutant ($p$) emitted (Thomson et al., 2016) as shown in Eq. (1).



$$EF_{fuel,p} = \frac{[p]}{[CO_2]\frac{M_C}{M_{CO_2}}+[CO]\frac{M_C}{M_{CO}}+[OC]+[BC]} w_c \qquad (1)$$

In Eq. (1), $w_c$ (g/kg-fuel) represents the measured effective fuel carbon fraction in dry basis, $M_C$, $M_{CO2}$ and $M_{CO}$ represent the molecular weight of carbon, $CO_2$ and $CO$, respectively. Energy-based emission factors ($EF_{energy}$, g/MJ) and brick-based

5  emission factors ($EF_{brick}$, g/kg-brick) are calculated using $EF_{fuel}$, the measured effective fuel heating value in dry basis ($w_f$, MJ/kg-fuel), and the specific energy consumption (SEC, MJ/kg-brick), respectively, as shown in Eqs. (2) and (3).

$$EF_{energy,p} = EF_{fuel,p} w_f{}^{-1} \qquad (2)$$

$$EF_{brick,p} = EF_{energy,p} SEC \qquad (3)$$

In Eq. (3), SEC is calculated by multiplying the fuel mass consumption rate (kg-fuel/day) by $w_f$ and dividing by the brick production rate (kg-bricks/day).

The second technique used to sample the kilns was based on the tracer ratio method in which the emission rate of the targeted

source is obtained by simultaneously measuring in real-time the above-background concentrations of the species of interest and of a selected gas tracer with a known release rate that is co-located at the emission source (Lamb et al., 1995). This method is based on the fundamental assumption that a relatively unreactive mixture of gases emitted from a common location experiences a quasi-perfect co-dispersion and equivalent dilution through the atmosphere. The source's emission rate ($ER$, l/s, standard conditions) can be estimated using the relationship between above-background concentrations of the species $p$ emitted

and the tracer $C_t$ multiplied by the known tracer's release flow rate $R_t$ (l/s, standard conditions) as shown in Eq. (4):

$$ER_p = \frac{[p]}{[C_t]} R_t \qquad (4)$$

Using the scaffold built for the measurements, the tracer was released at a constant rate close to the top of the kiln so that the

kiln's emissions and the released tracer were simultaneously transported downwind and measured by the instruments on-board the Aerodyne Mobile laboratory (AML). For this study nitrous oxide ($N_2O$) and ethyl acetate ($C_4H_8O_2$) were used as tracer gases for the measurement of the $ER$ due to their low atmospheric reactivity and the ability of the AML to measure their concentrations very accurately and with high sensitivity. The $C_4H_8O_2$ emission tracer was generated by bubbling air through a bottle containing the compound. While it was co-located with the known $N_2O$ emission, its direct release rate was uncertain.

Thus, only the $N_2O$ tracer was used to quantify brick kiln emission rates. The $C_4H_8O_2$ served as an auxiliary tracer identified when the AML was downwind of plumes from the kiln of interest, rather than from other sources in the area. Furthermore, it





independently diagnosed the tracer plume characteristics directly with the instrumentation used to measure VOCs of interests, as described below.

The AML incorporates real-time data acquisition and data display capabilities so that *in-situ* decisions by the investigators can be made to move the laboratory in and out of the emission plumes that are identified by tracer detection. This is a key element for the successful application of this technique since the dilution and advection of the kiln emissions are dictated by local meteorological conditions that can vary in short time scales. The mobile laboratory was typically positioned between 20-100 m from the kiln during the tracer ratio measurements. The tracer ratio method allows the unequivocally identification of emission plumes from the targeted kiln at various time periods of its operation process; this in turn allows to further apply the mass carbon method to the identified plumes following Eqs. (1-3) to obtain fuel, energy, and brick-based emission factors that can be compared to those obtained with the filter-based technique.

The instrumentation on-board the AML included a soot particle aerosol mass spectrometer (SP-AMS) developed by Aerodyne Research Inc. (Onasch et al., 2012), which measured BC and OC using laser-induced incandescence of absorbing soot particles to vaporize both the coatings and BC cores of exhaust soot particles within the ionization region of the AMS (Dallman et al., 2014). The SP-AMS also measured other inorganic PM components including nitrates, sulfates, ammonium, and chlorides corresponding to a particle size range of 50 – 600 nm. In this study, we refer to PM emission factors obtained with the AML as the sum of BC, OC and inorganic components simultaneously measured with the SP-AMS.

The AML measured $N_2O$, $CH_4$, $C_2H_6$, $SO_2$, CO and acetylene ($C_2H_2$) using Tunable Infrared Laser Differential Absorption Spectrometers (TILDAS); $NO/NO_y$ were measured using a Thermo Electron 42i chemiluminescent detector modified for fast-response; a LiCor 6262 Non-Dispersive Infrared (NDIR) instrument measured $CO_2$; and a Proton Transfer Reaction Mass Spectrometry (PTR-MS) using $H_3O^+$ as the ionization reagent was operated in multiple ion detection mode to measure selected VOCs (Rogers et al., 2006). Species measured with the PTR-MS included methanol, acetonitrile, acetaldehyde, acetone, benzene, toluene, acetic acid, ethyl acetate, C2-benzenes (sum of $C_8H_{10}$ isomers: xylenes, ethylbenzene, and benzaldehyde), and C3-benzenes (sum of $C_9H_{12}$ isomers and $C_8H_8O$ isomers). Calibrations of these instruments were checked using certified gas standards. Other instruments on-board the mobile laboratory included a global positioning system (GPS), a sonic anemometer, and a video camera. Further details on the AML instruments detection limits and sensitivities are presented in Table SM5 of the Supplemental Material document.

## 3 Results

The average fuel-based emission factors (g/kg-fuel) obtained with both the sampling probe and tracer ratio techniques are shown in Table 2. The table also shows the modified combustion efficiency (MCE) that is obtained as the ratio of $CO_2$ to ($CO_2$



+ CO) concentrations and thus is a useful indicator of the combustion efficiency. The corresponding brick- and energy-based emission factors for the three kilns are shown in Tables SM6 and SM7, respectively, in the Supplemental Material document. Table 2 also shows the average emission rates (g/min) obtained for the three kilns with the tracer ratio technique.

As shown in Table 2, the relative variability of emission rates is much higher compared to the variability of fuel-based emission factors. Time-based emission rates are highly variable particularly during the burning stage because they strongly depend on the fuel-feeding practices including the amount and type of fuel used, as well as the operator's decision of when to add fuel. The lower variability of the fuel-based emission factors compared to emission rates indicates that the normalization of the emissions of combustion by-products effectively takes into account the variations in the thermal energy employed in the
cooking process. In addition, since estimations of integrated emissions burden using emission rates depend on the total brick production time, emission rates are not a good indicator to compare the environmental performance of the kilns. However, emission rates can be useful during the development of emissions inventories as inputs in air quality models to better understand the time-based chemical evolution of the emitted species at local and urban scales.

A comparison of temporal profiles of CO, BC, and OC fuel-based emission factors for the traditional-fixed kiln between the two techniques is shown in Fig. 2. Comparisons of the temporal profiles for all measured pollutants are shown in Fig. SM1, SM2 and SM3 for the MK2, traditional-campaign, and traditional-fixed kilns, respectively, in the Supplemental Material document. The results show that in general both techniques capture comparable temporal profiles of the kiln emissions while the magnitudes of the emission factors are remarkably similar. As the fuels used in the three kilns were mostly wood, the
resulting identities of VOCs emitted are similar to those from biomass burning. Furthermore, the temporal profiles shown in Figs. SM1-SM3 indicate that high levels of VOCs can be emitted not only during the burning stage of the brick cooking process but also during the smoldering and cooling stages. Measurements in this pilot study focused primarily on the burning stages and only included partial periods of the smoldering and cooling stages. Therefore, a complete characterization of VOC emissions for brick kilns would require the measurement of the full brick-cooking period.

The data from the tracer ratio technique show that there is large short-term variability of the emission factors for both gaseous and particulate pollutants during the burning stage of the cooking process; this variability is only partially captured by the filter-based sampling probe technique. On the other hand, whereas the sampling probe technique continuously measures the kiln's emissions in ~1-hour intervals, the tracer technique strongly depends on the capability to position the mobile laboratory
downwind at a distance ranging from approximately 20-100 m from the kiln, depending on wind speed, and is not feasible during stagnant wind conditions. Thus, with the tracer ratio technique there may be unavoidable gaps in the data needed to fully characterize the kiln's emissions for the entire process. As shown in Fig. SM1, the MK2 kiln presented the largest data gaps with the tracer ratio technique and thus, the obtained emission factors in Table 2 may not represent the complete brick production process with this technique. Therefore, in our subsequent discussions and for the comparison of particulate emission





factors we have used the results obtained with the sampling probe technique because all the available comparable studies with PM data used filter-based measurements. This includes emission factors for $PM_{2.5}$, BC, OC, and all the inorganic and ionic PM components. However, in this study all sampled VOCs, $SO_2$, $NO_x$, and $CH_4$ were obtained using only the tracer ratio technique and thus these results are used in the comparisons.

Major components of $PM_{2.5}$ for the MK2 and the traditional-campaign kilns are distinctively different than for the traditional-fixed kiln. As described, the two former kilns belong to a different brick production community (El Refugio) and used similar mix of fuels and batches of clay, whereas the traditional-fixed kiln used mostly avocado wood and a different batch of clay as it is located in a different community (Abasolo). Total carbon corresponded to 9.3, 12.5, 51.1 % in mass of $PM_{2.5}$ for the MK2,

the traditional-campaign, and the traditional-fixed kilns, respectively. Correspondingly, BC accounted for 7.8, 6.0, and 40.5 % of PM2.5 for the three brick kilns. Chloride (31.9 %, 42.4%), ammonium (12.7 %, 20.4%), potassium (9.8 %, 2.5%), and sulfate (7.0 %, 2.0%) were the predominant mass components in $PM_{2.5}$ for the MK2 and the traditional-campaign kilns, respectively, whereas the sum of these four components amounted to only 8.9% in mass of $PM_{2.5}$ for the traditional-fixed kiln.

The measured ionic contents are quite high for the MK2 and the traditional-campaign kilns, the sum is greater than the BC + OC content. This indicates that either the ash content of the fuels is quite high or that these non-combustible inorganics are abundant in the brick material. The chloride content is especially elevated, which is often seen when trash containing plastics are burned. In our measurements we controlled the fuels types feed to the kilns and no chlorinated materials were used. Since the clay used for these two kilns was obtained locally in the same brick production community, it is possible that it may be

already contaminated with PM deposition resulted from continued trash-burning practices during brick production over the years. This suggests that environmental and health impacts of brick production can be further persistent even after the banning of trash burning practices.

Fluorides, bromides and other halogens are not typically high in ambient filter-based PM samples but they may be present in

trace amounts in clay. Previous work has shown that fluorides from brick kilns can have adverse effects on vegetation and crops (Ahmand et al., 2012). Emission factors of particulate fluorides in this study were small (1.1-2.2 x $10^{-3}$ g/kg-fuel), suggesting that it was not present in large amounts in the raw brick materials. None of the wood used during the burning stage had paint or solvents on it, thus ruling out possible contributions of halogens or metals from wood fuels. Nevertheless, it has been reported that these materials can be used as part of wood waste products utilized as fuels by brick producers in Mexico

(CIATEC, 2015).

It should be noted that during these measurements both methane and ethane emissions were quantified aboard the AML. The ethane measurement is an important complement to methane because it is a marker for non-biogenic methane emissions. Interestingly, the mass ratio of ethane to methane was consistently 0.06-0.075 between the three brick kiln types despite the



substantial variation of the $CH_4$ emission ratio. This indicates that the ethane production is strongly linked to the methane production, and the ratio is not strongly dependent on the brick kiln operation.

## 4 Discussions

### 4.1 Brick cooking process

The physical and chemical changes occurring in the bricks during the cooking process are associated with the burning, smoldering and cooling stages, which are in turn determined by changes in thermal energy transfer rates within the kiln, and are closely related to the final quality of the cooked bricks. In describing the brick cooking process, we define the burning stage as the time passed since the firing starts until the feeding of fuel is stopped, the smoldering stage as the time when the maximum temperature at the top of the kiln is reached minus the burning time, and the cooling stage as the time when the temperature at the bottom of the kiln reaches a stable minimum minus smoldering time. The temporal profiles of temperature at the lower, middle, and upper levels of the kilns and the brick cooking stages are shown in Fig. 3. The corresponding rates of heating and cooling are obtained as the time derivatives of the temperature profiles.

The data show that the cooking of bricks results from vertical transfer of thermal energy inside the kiln starting from the beginning of the burning stage when temperatures at the bottom layers rise quickly with very high heating rates. In general, higher temperatures are reached inside the traditional-fixed kiln, followed by the traditional-campaign and the MK2 kilns. The bricks located in the middle and upper layers of the kiln start their cooking process only after sufficient thermal energy is transferred from the bottom layer. Interestingly, in the case of the MK2 and the traditional-campaign kilns this can occur during the burning stage, but for the traditional-fixed kiln the cooking of bricks at the middle and upper layers occur only during the smoldering and cooling stages.

During the burning stage at the bottom of the kiln, the heating rate is much higher and smoother in the case of the traditional-fixed kiln compared to the traditional-campaign kiln, whereas the MK2 kiln shows highly variable but overall decreasing heating rates. This critical difference in the heating process at the burning stage is likely due to the physical arrangement of bricks and the design of the kiln. The traditional-fixed kiln seems to be particularly efficient in its vertical thermal energy transfer inside the kiln as temperatures in the middle and upper levels reach similarly high values (and at comparable heating rates) as those at the bottom even after the burning stage has finished.

The primary effect of the initial period of the burning stage is to remove all the remaining moisture from the bricks. At the beginning of the process this is done only at the bottom layers as temperatures do not reach high values in the middle and upper layers until much later. Once the moisture is removed and the temperatures continue rising, the carbonaceous organic material contained in the clay is removed by combustion. The raw materials for the three kilns are comparable in mass and



type of clay used, but the traditional-campaign and the MK2 kilns use about 3.5 wt.% of manure whereas the traditional-fixed kiln use 8 wt.% of sawdust (see Table 1). These materials are additives that the brick producers use during the clay preparation process, mixing them with water and crushing them until the mixture is ready for molding. These organic additives effectively act as internal fuel during the brick cooking process and affect the quality and mechanical condition of the bricks (Martínez

and Jiménez, 2014).

As temperatures continue to rise, the hydroxyl groups that are combined with the chemical compounds forming the clay begin the process of dehydroxylation, which effectively releases water and other volatile compounds at about 450 ℃ (Osornio-Rubio et al., 2016). Figure 3 shows that the traditional-fixed kiln reaches dehydroxylation much faster than the MK2 and traditional-

campaign kilns. At about 573 ℃ ($T_v$ in Fig. 3) the silica contained in the clay changes its α–quartz structure to a β-quartz structure, effectively expanding the volume of the clay (Heaney and Veblen, 1991). If the temperatures throughout the brick are not homogenous around $T_v$, cracks in the brick can form due to the mechanical stress of different volume expansion (Weyant et al., 2016).

Above Tv the clay begins the actual vitrification process in which clay particles melt to form a glassy bond, ultimately giving strength to the brick during the cooling stage. Brick producers have learned by experience the importance of not extending the vitrification process more than what is needed as overheating may distort the shapes of the bricks. Similarly, if the vitrification is not achieved homogenously within the brick, the mechanical resistance and thus the quality of the final product will be smaller. The time that the bricks are exposed to temperatures above $T_v$ is 1.9 and 1.2 times larger for the traditional-campaign

kiln compared to the traditional-fixed and MK2 kilns, respectively. Therefore, of the three kilns the traditional-fixed kiln exposes the bricks to temperatures above $T_v$ for much shorter periods of time. In addition, the time-integrals of the temperature profiles above $T_v$ of the traditional-fixed kiln are at least half the magnitude of the corresponding time-integrals for the traditional-campaign and MK2 kilns, indicating that much less thermal energy is transferred inside the traditional-fixed kiln for vitrification.

**4.2 Comparison among sampled kilns**

The environmental performance of the brick kilns can be assessed in terms of the relative magnitude of the emission factors during the brick production process. The use of fuel-based emission factors to compare brick kilns performance is adequate when similar fuels are used among different kilns and when bricks have similar physical characteristics. In contrast, energy-based emission factors are adequate comparison indicators when fuels types are substantially different because they take

directly into account the effective heating value of the fuels employed. Brick-based emission factors are adequate comparison indicators between kilns when the mass and size of the bricks produced are substantially different. Nevertheless, regardless of the type of emission factor used, an integrated assessment of the brick kilns performance should also incorporate other



parameters, such as energy efficiency, fuel consumption, combustion efficiency, production time, and the quality of bricks produced, among others.

Figure 4 shows an inter-comparison of the relative performance of the three sampled kilns along with the specific energy consumption, fuel consumption, modified combustion efficiency, and measured brick's mechanical resistance as a surrogate for bricks' quality. In order to compare the relative environmental performance of the three kilns, we have normalized the fuel-based emission factors with the corresponding average of the three kilns for each pollutant in Fig. 4. Regardless of the base (fuel mass, brick mass, or energy) employed, the normalization effectively allows the simultaneous comparison of emissions factors for multiple pollutants that differ by orders of magnitude while providing information on their relative magnitudes.

The results show that the traditional-fixed kiln had lower modified combustion efficiency, lower fuel (wood) consumption, and slightly higher specific energy consumption compared to the other two kilns. The results of the measured mechanical resistance of the bricks produced are shown in Table SM8 of the Supplemental Material document. The traditional-fixed kiln also produced bricks with an average mechanical resistance almost half of that compared to the traditional-campaign kiln, in agreement with the much higher time-integral of the temperature profile above $T_v$ for the traditional-campaign kiln and suggesting a more efficient vitrification process. Thus, although bricks from the traditional-fixed and MK2 kilns complied with the Mexican standard, the much higher mechanical resistance in the traditional-campaign kiln indicates that its bricks were produced with higher quality.

Low combustion efficiency is related to higher pollutant emissions produced during incomplete combustion. The traditional-fixed kiln had the highest emissions factors for CO, BC, as well as $CH_4$, $C_2H_6$, $CH_3OH$, $C_2H_3N$, and $C_2H_4O$ but emitted substantially smaller inorganic PM components. Conversely, CO, BC and OC emission factors were much smaller for the MK2 kiln compared to the traditional-campaign and traditional-fixed kilns, but had the highest inorganic PM components. Although the latter minimally contribute in mass to the overall emissions, ionic species may be important contributors to chemical processes in the atmosphere involving wet deposition. The measured SEC values were similar for the three kilns, with 10% variation among them, because the fuels used had similar heating values. In addition, since the combustion efficiency for the MK2 and the traditional-campaign kilns are somewhat similar in magnitude, the results indicate that the traditional-campaign kiln produced bricks of much higher quality while performing more efficiently in energy consumption and combustion efficiency than the other kilns.

## 4.3 Comparison with other studies

Very few studies are available on the chemical characteristics of emission factors for brick kilns. Previous work by Christian et al. (2010) includes measurements of multiple gases and PM composition for three traditional-fixed kilns in Mexico that used wood waste products as fuel. Of the five types of kiln designs measured by Rajarathnam et al. (2014) and Weyant et al. (2014)

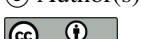



in India and Vietnam, only the down-draft kiln type used wood as fuel while the rest used mostly coal. Both the "zig-zag" and clamp kilns measured by Stockwell et al. (2016) in Nepal also used coal as fuel. Jayarathne et al. (2017) recently reported the particle-phase results of the same kilns measured by Stockwell et al. (2016). Of these studies, Stockwell et al. (2016) and Christian et al. (2010) report fuel-based energy factors whereas Rajarathnam et al. (2014) and Weyant et al. (2014) report

energy-based emission factors, allowing a proper inter-comparison with our results. Tables 3 and 4 show a comparison of the energy-based and fuel-based emission factors, respectively, with those obtained in other studies.

Table 3 shows that the specific energy consumption for brick kilns using coal as fuel in the studies of Rajarathnam et al., (2014) and Weyant et al., (2014) are much smaller than for those measured in this study, due to the much higher energy density

content of coal versus wood. $SO_2$ emission factors for coal-firing kilns are higher than those of wood-firing kilns; coal having larger sulfur content than wood. Nevertheless, the major difference between emissions factors among the kilns seems to be caused by the kiln design. The improved designs for the zig-zag and vertical shaft kilns are related to substantially smaller emission factors than the other kilns, indicating large environmental benefits by the use of more efficient brick kiln technologies. Thus, addressing the complex economic, social, and technical barriers surrounding the adoption of more efficient

technologies can produce substantial environmental and health benefits.

The emission factors in this study are closer to the values reported for the down-draft kiln by Rajarathnam et al. (2014) and Weyant et al. (2014) and to the results by Christian et al. (2010) due to similarities in kiln designs and fuels (wood) employed. However, there are differences in the emission factors that suggest substantial inter-variability of emissions even when fuels

and kilns designs are similar. The average BC and OC emission factors obtained in this study for the traditional-fixed kiln of 0.54 and 0.14 g/kg-fuel, respectively, are within the lower range of the values reported by Christian et al. (2010), whereas the corresponding BC and OC energy-based emission factors are 2-12 times lower than those reported by Weyant et al. (2014) for the down-draft wood-fueled kiln. In the study of Weyant et al. (2014), the sample streams were diluted and cooled before measuring whereas our filter-based measurements were not diluted. Similarly, in the study of Stockwell et al., (2016) the

emissions were sampled downwind of the stack after natural dilution and cooling. As gas-to-particle mass transfer processes are likely to occur under strong temperature gradients, different sampling techniques can further contribute to observed differences.

In our study the BC/OC ratios were 5.2, 0.9, and 3.8 for the MK2, traditional-campaign, and traditional-fixed kilns,

respectively; whereas the corresponding BC/OC ratios in Christian et al. (2010) ranged from 5.29 to 8.15. Methane, methanol, and acetic acid fuel-based emission factors for the traditional-fixed kiln are 3-5, 2-8, and 5 times higher, respectively, than those reported by Christian et al. (2010). These higher emission factors are consistent with the lower average MCE of 0.910 obtained in this study compared to the average MCE of 0.968 for the traditional-fixed kiln sampled by Christian et al. (2010).



As a comparison, Stockwell et al. (2016) reports a much higher average MCE value of 0.994 for the zig-zag coal-fueled brick kiln sampled.

Overall, the comparison of the results in this study with the available literature reports indicate that there is substantial variability among brick kiln designs and fuel types. Due to the small sampling size, it is not possible to distinguish from the data the contribution of fuel types and kiln design to the overall variability of emissions during brick production. Therefore, although both the traditional-campaign and traditional-fixed kilns are widely used in Mexico, caution should be taken into generalizing the results to other brick production regions with different fuels and operation practices. Nevertheless, since the number of studies with chemical composition of brick kiln emissions is so small, the results of this study represent valuable additions to the current literature.

## 5 Conclusions

Despite the widespread use of brick kilns in Mexico and other Latin American countries, there have been very few studies on their emission characteristics. An important part of the brick production in Mexico is still done by using traditional brick kilns that are operated with artisanal methods and thus the individual kiln's performance depends on the producer's operation skills, kiln design, and available materials and fuels. This diversity in operating conditions can result in large intra-variability on the pollutant emissions characteristics from brick kilns even when using similar designs and fuels. Therefore, there is a need for additional emissions measurements from brick production to better constrain the uncertainties of emissions estimates and mitigate their environmental and human health impacts. Since the tracer ratio method is not limited by mass saturation constrains, the results from this pilot project suggest that the tracer technique can be an alternative option to the filter-based sampling probe technique in understanding the temporal profile of the chemical composition of brick kilns emissions.

The results of this study showed that a well-designed and operated MK2 kiln produced lower $PM_{2.5}$, BC, CO, and OC emission factors, the traditional-campaign kiln overall had the lowest sampled VOCs emission factors, whereas the traditional-fixed kiln had the lowest inorganic PM component emission factors. However, we have shown that non-environmental parameters can be used to quantitatively evaluate the performance of brick kilns. The traditional-campaign kiln had good energy efficiency performance and produced bricks with the highest quality, likely due to a better vitrification process. The MK2 kiln had a short cooking time and similar energy consumption to the traditional fixed and campaign kilns. Despite its higher internal temperatures, smaller fuel consumption, and shorter burning time, the traditional-fixed kiln produced lower quality bricks and with overall high emissions of combustion products. As both energy-efficient and low-emissions brick kilns are needed to mitigate the impacts from these sources, further studies should address the benefits of potential upgrades in the mechanical design of kilns to further improve their fuel consumption and energy efficiency.



Disclaimers:

The authors declare that they have no conflict of interest.

**Acknowledgments**

The SLCF-2013 Mexico measurement field campaign was coordinated by the Molina Center for Energy and the Environment under UNEP Contract GFL-4C58. MZ and LTM acknowledge additional support from NSF Award 1560494. The authors would like to thank the local brick producers from the Asociación de Productores de Barro y Arcilla del Refugio A. C., and the Productores de Ladrillo de Abasolo for their participation in this study. Special thanks to the Instituto de Ecología del Estado de Guanajuato (IEEG), Francisco Guardado from the Instituto Nacional de Ecología y Cambio Climático (INECC),

and Carlos Frias Mejía from the Asociación de Productores de Barro y Arcilla del Refugio A. C. for logistical support.

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





Table 1. Summary of the characteristics of kilns sampled.

| Parameter | Traditional-fixed kiln | Traditional-campaign kiln | MK2 kiln |
|---|---|---|---|
| Burning time[a] [hr] | 3.8 | 20.5 | 17.6 |
| Soaking time[b] [hr] | 17.1 | 14.4 | 19.2 |
| Cooling time[c] [hr] | 10.2 | 23.1 | 12.5 |
| Total raw bricks [pc] | 21780 | 9898 | 5135 |
| Bricks rejected [%][d] | 0.20 | 1.8 | 2.8 |
| Mass of raw brick [kg] | 3.59 ± 0.05 | 4.44 ± 0.16 | 4.55 ± 0.16 |
| Mass of cooked brick [kg] | 3.03 ± 0.04 | 3.95 ± 0.25 | 4.11 ± 0.16 |
| Moisture content in raw bricks [wt.%] | 15.6 | 11.2 | 9.6 |
| Carbon content in raw brick [wt.%] | 0.86 | 1.28 | 1.28 |
| Carbon content in cooked brick [wt.%] | 0.13 | 0.11 | 0.11 |
| Raw materials [wt.%] | clay (92), sawdust (8) | clay (96.4), manure (3.6) | clay (96.6), manure (3.4) |
| Fuels[e] | wood, diesel, sawdust | wood, manure | wood, manure |

[a] Time passed since the firing starts until the fuel feeding is stopped.

[b] Time when the maximum temperature at the top of the kiln is reached minus burning time.

[c] Time when the temperature at the bottom of the kiln reaches a stable minimum minus smoldering time. It takes about 48 hours for a kiln to homogenously cool off back to ambient temperature.

[d] Percentage of bricks either broken or fractured after burning, thus rejected for sale.

[e] See Supplemental Material document for specific types, quantities, and chemical composition of fuels.



Table 2. Average modified combustion efficiency (MCE), fuel-based emission factors EF [g/kg fuel], and emission rates ER [g/min] obtained with the sampling probe (SP) and tracer ratio (AML) techniques.[1]

| | MK2 | | | Traditional-campaign | | | Traditional-fixed | | |
|---|---|---|---|---|---|---|---|---|---|
| | SP | AML | | SP | AML | | SP | AML | |
| | EF-fuel | EF-fuel | ER | EF-fuel | EF-fuel | ER | EF-fuel | EF-fuel | ER |
| MCE | 0.96 | 0.94 | | 0.95 | 0.94 | | 0.91 | 0.92 | |
| | (0.02) | (0.04) | | (0.02) | (0.03) | | (0.02) | (0.02) | |
| $CO_2$ | 1583 | 1595 | | 1527 | 1597 | | 1668 | 1658 | |
| | (28) | (58) | | (28) | (54) | | (40) | (43) | |
| CO | 44.4 | 65.4 | 270.7 | 50.5 | 65.3 | 553.7 | 105.2 | 105.3 | 8500.2 |
| | (18) | (526) | (902) | (17) | (43) | (1040) | (24) | (36) | (9588) |
| TOC[2] | 2.0 | | | 5.0 | | | 14.6 | | |
| | (2) | | | (4) | | | (2) | | |
| $CH_4$ | | 2.39 | 11.90 | | 3.34 | 28.0 | | 5.92 | 551 |
| | | (2.6) | (28) | | (2.9) | (53) | | (2.2) | (699) |
| NO | | 1.02 | 4.3 | | 1.05 | 13.8 | | 0.76 | 43.4 |
| | | (0.9) | (8) | | (2.1) | (71) | | (0.3) | (31) |
| $NO_2$ | | 1.7 | 7.4 | | 0.93 | 7.8 | | 1.01 | 53.8 |
| | | (1.8) | (18) | | (1.4) | (27) | | (0.6) | (36) |
| $SO_2$ | | 1.0 | 3.6 | | 0.27 | 1.1 | | 0.13 | 8.7 |
| | | (1.4) | (8) | | (0.3) | (2) | | (0.1) | (9) |
| $PM_{2.5}$[3] | 1.94 | 1.66 | 3.9 | 4.62 | 2.28 | 17.5 | 1.32 | 1.26 | 171.9 |
| | (0.6) | (0.8) | (2) | (4.3) | (1.8) | (15) | (1.3) | (2.2) | (152) |
| BC | 0.15 | 0.67 | 1.6 | 0.28 | 0.73 | 3.4 | 0.54 | 1.03 | 149.4 |
| | (0.1) | (0.5) | (3) | (0.2) | (0.6) | (5) | (0.8) | (2.2) | (377) |
| OC | 0.03 | 0.52 | 1.5 | 0.3 | 1.18 | 12.6 | 0.14 | 0.18 | 19.5 |
| | (0.03) | (0.6) | (5) | (0.7) | (1.7) | (38) | (0.1) | (0.2) | (31) |
| Fullerene ($\times 10^{-3}$) | | 31.0 | 84.5 | | 26.8 | 150.1 | | 8.1 | 912.4 |
| | | (39) | (183) | | (30) | (252) | | (12) | (1820) |
| Ammonium ($\times 10^{-3}$) | 246.3 | 96.8 | 155.2 | 942.2 | 66.7 | 240.6 | 2.9 | 6.6 | 312.5 |
| | (105) | (64) | (225) | (1068) | (69) | (442) | (2) | (5) | (349) |
| Nitrate ($\times 10^{-3}$) | 1.4 | 23.4 | 68.4 | 10.7 | 12.3 | 69.7 | 3.4 | 4.8 | 393.6 |
| | (1) | (33) | (173) | (17) | (20) | (197) | (2) | (6) | (691) |
| Sulfate ($\times 10^{-3}$) | 135.5 | 121.3 | 315.1 | 91.9 | 68.8 | 302.1 | 48.8 | 21.6 | 1721.7 |
| | (96) | (203) | (678) | (36) | (104) | (620) | (59) | (19) | (2240) |
| Chloride ($\times 10^{-3}$) | 617.2 | 234.1 | 305.2 | 1956.3 | 226.8 | 936.8 | 21.7 | 10.5 | 649.7 |
| | (200) | (153) | (397) | (2095) | (227) | (1766) | (14) | (4) | (624) |
| Sodium ($\times 10^{-3}$) | 27.2 | | | 21.9 | | | 15.3 | | |
| | (15) | | | (9) | | | (12) | | |
| Magnesium ($\times 10^{-3}$) | 0.67 | | | 1.34 | | | 1.11 | | |
| | (0.4) | | | (0.8) | | | (0.8) | | |
| Potassium ($\times 10^{-3}$) | 189.3 | | | 114.7 | | | 44.0 | | |
| | (146) | | | (74) | | | (65) | | |
| Calcium ($\times 10^{-3}$) | 5.5 | | | 7.9 | | | 5.3 | | |




| | | | | | | |
|---|---|---|---|---|---|---|
| | (4) | | (4) | | (3) | |
| Fluoride (x10⁻³) | 1.1 | | 2.2 | | 1.2 | |
| | (1) | | (2) | | (1) | |
| Chloride (x10⁻³) | 617.2 | | 1956.3 | | 21.7 | |
| | (200) | | (2095) | | (14) | |
| Bromide (x10⁻³) | 5.0 | | 23.6 | | 1.0 | |
| | (2) | | (38) | | (1) | |
| Ethane | 0.15 | 0.6 | 0.21 | 1.1 | 0.44 | 30.0 |
| | (0.2) | (1) | (0.2) | (3) | (0.1) | (29) |
| Methanol | 1.99 | 18.7 | 1.19 | 5.1 | 3.25 | 185.3 |
| | (2) | (119) | (2.3) | (18) | (1.2) | (151) |
| Acetonitrile | 0.24 | 1.1 | 0.15 | 0.7 | 0.46 | 30.7 |
| | (0.2) | (2) | (0.1) | (1) | (0.2) | (34) |
| Acetaldehyde | 1.13 | 5.8 | 0.54 | 2.2 | 2.18 | 146.9 |
| | (1.2) | (12) | (0.4) | (3) | (0.5) | (129) |
| Acetone | 1.28 | 6.3 | 0.61 | 2.6 | 0.91 | 70.2 |
| | (1.5) | (12) | (1.9) | (14) | (0.3) | (79) |
| Acetic acid | 2.64 | 11.6 | 0.89 | 2.0 | 1.04 | 38.2 |
| | (3.1) | (22) | (2.6) | (4) | (0.8) | (20) |
| Benzene | 0.84 | 3.8 | 0.66 | 3.4 | 0.5 | 58.3 |
| | (0.9) | (7) | (0.7) | (6) | (0.3) | (83) |
| Toluene | 0.93 | 5.3 | 0.42 | 1.9 | 0.28 | 20.0 |
| | (0.8) | (11) | (0.9) | (7) | (0.2) | (20) |
| C2Benzenes | 1.01 | 5.6 | 0.54 | 2.1 | 0.19 | 11.5 |
| | (1.1) | (11) | (1.5) | (10) | (0.1) | (11) |
| C3Benzenes | 0.86 | 4.7 | 0.45 | 1.7 | 0.13 | 7.4 |
| | (1.) | (10) | (1.2) | (10) | (0.1) | (7) |

[1] Emission factors obtained with the filter technique represent 1-hr continuous measurements of the brick production process, whereas those obtained with the tracer technique represent sporadic sampling times of a few tens to hundreds of seconds. See text and supplemental material for sampling details. Values in parenthesis are 1 standard deviation. See the Supplemental Material document for the corresponding energy and kg-brick based emission factors.

[2] Total organic carbon measured as methane equivalent.

[3] PM mass and its components from the AML results represent PM in the range 50-600 nm.





Table 3. Comparison of energy-based emission factors (g/MJ) measured in this study with other studies.

| kiln type | This study | | | Rajarathnam et al., (2014) and Weyant et al., (2014)[a] | | | | | | |
| | MK2 | Traditional campaign | Traditional fixed | Fixed chimney Bull's trench | Natural draft zig-zag | Forced draft zig Zag | Vertical shaft | Down-Draft | Vertical shaft[b] | Tunnel[b] |
|---|---|---|---|---|---|---|---|---|---|---|
| Fuels | Wood | Wood | Wood, diesel, sawdust | Coal, wood, others | Coal, wood, others | Coal | Coal | Wood | Coal | Coal |
| SEC[c] | 2.07 | 2.16 | 2.22 | 1.1 - 1.46 | 1.02 - 1.21 | 0.95 - 1.11 | 0.95 | 2.91 | 0.54 | 1.47 |
| $PM_{2.5}$ | 0.11 (0.04) | 0.27 (0.2) | 0.07 (0.1) | 0.07-0.23 | 0.03-0.19 | 0.03-0.05 | 0.053 | 0.17 | 0.16 | 0.163 |
| BC | 0.01 (0.01) | 0.02 (0.01) | 0.03 (0.04) | 0.08-0.18 | 0.008-0.029 | 0.004-0.019 | 0.002 | 0.06 | 0.002 | 0.001 |
| OC | 0.002 (0.002) | 0.018 (0.04) | 0.007 (0.006) | 0.004-0.008 | 0.00-0.012 | 0.00-0.012 | 0.030 | 0.024 | 0.00 | 0.00 |
| $SO_2$ | 0.058 (0.08) | 0.016 (0.02) | 0.007 (0.003) | 0.39 (0.92) | 0.06 (1.52) | 0.23 (1.00) | 0.11 (0.19) | <0.1 (0.04) | 1.78 (0.01) | 0.49 (0.03) |
| CO | 2.57 (1) | 2.94 (1) | 5.56 (1.3) | 2.96 (0.91) | 0.32 (0.97) | 1.96 (0.76) | 4.39 (0.39) | 5.17 (0.04) | 2.93 (0.12) | 1.56 (0.26) |
| $CO_2$ | 91.4 (2) | 88.9 (2) | 88.1 (2) | 86.8-108.2 | 96.0-102.7 | 88.1-99.6 | 83 | 93.3 | 110.4 | 111.2 |

[a] Emission factors for $CO_2$, OC, BC (obtained as EC), and $PM_{2.5}$ are reported by Weyant et al., (2014); CO and $SO_2$ are reported by Rajarathnam et al., (2014). Numbers in parenthesis for Rajarathnam et al., (2014) and represent the ratio of standard deviation to the mean whereas in this study the values in parenthesis represent the 1 standard variation.

[b] These two kilns were measured in Vietnam, whereas the rest of kilns in Rajarathnam et al., (2014) and Weyant et al., (2014) were sampled in India.

[c] SEC indicates Specific Energy Consumption in MJ/kg-brick.



Table 4. Comparison of fuel-based emission factors (g/kg-fuel) measured in this study with other studies.

| kiln type | This study[a] | | | Stockwell et al., (2016), Jayarathne et al., (2017), Nepal | | Christian et al., (2010), Mexico |
|---|---|---|---|---|---|---|
| | MK2 | Traditional-campaign | Traditional-fixed | Clamp | Forced draft zig Zag | Traditional-fixed |
| Fuels | Wood | Wood | Wood, sawdust | Coal, hardwood | Coal, bagasse | Sawdust, wood waste |
| $PM_{2.5}$ | 1.94 (0.6) | 4.62 (4.3) | 1.32 (1.3) | 10.7 (1.6) | 15.1 (3.7) | 1.2 – 2.0[b] |
| BC | 0.15 (0.1) | 0.28 (0.2) | 0.54 (0.8) | 0.0172 | 0.112 | 0.596-1.5 |
| OC | 0.03 (0.03) | 0.3 (0.7) | 0.14 (0.1) | 6.74 | 1.05 | 0.073-0.283 |
| $SO_2$ | 1. (1.4) | 0.27 (0.3) | 0.13 (0.1) | 13 | 12.7 | |
| CO | 44.4 (17.7) | 50.5 (16.7) | 105.2 (24.3) | 70.9 | 10.1 | 25.7-55.7 |
| $CO_2$ | 1582 (28) | 1526 (28) | 1668 (40) | 2102 | 2620 | 1736-1787 |
| NO | 1.02 (0.9) | 1.05 (2.1) | 0.76 (0.3) | bdl | 1.28 | |
| $NO_2$ | 1.7 (1.8) | 0.93 (1.4) | 1.01 (0.6) | 0.297 | 0.0821 | |
| $CH_4$ | 2.39 (2.6) | 3.34 (2.9) | 5.92 (2.2) | 19.5 | 0.0873 | 1.13-2.16 |
| $C_2H_6$ | 0.15 (0.2) | 0.21 (0.2) | 0.44 (0.1) | 5.37 | 0.00206 | |
| $CH_3OH$ | 1.99 (2.) | 1.19 (2.3) | 3.25 (1.2) | 1.77 | 0.112 | 0.39-1.42 |
| $CH_3COOH$ | 2.64 (3.1) | 0.89 (2.6) | 1.04 (0.8) | 0.43 | 0.471 | 0.21 |
| $C_6H_6$ | 0.84 (0.9) | 0.66 (0.7) | 0.5 (0.3) | 1.68 | 0.00825 | |
| $C_6H_5CH_3$ | 0.93 (0.8) | 0.42 (0.9) | 0.28 (0.2) | 1.05 | 0.0028 | |
| $C_3H_6O$ | 1.28 (1.5) | 0.61 (1.9) | 0.91 (0.3) | - | 0.146 | |
| $C_2H_4O$ | 1.13 (1.2) | 0.54 (0.4) | 2.18 (0.5) | 0.0413 | 0.0694 | |

[a] Numbers in parenthesis represent 1 standard variation. Values in Crhistian et al, (2010) represent range of averages. "bdl" indicates below detection limit; "-" indicates concentrations were not greater than background.

5 [b] Estimated from measurements of OC, EC, metals, and ions (but not sulfate).





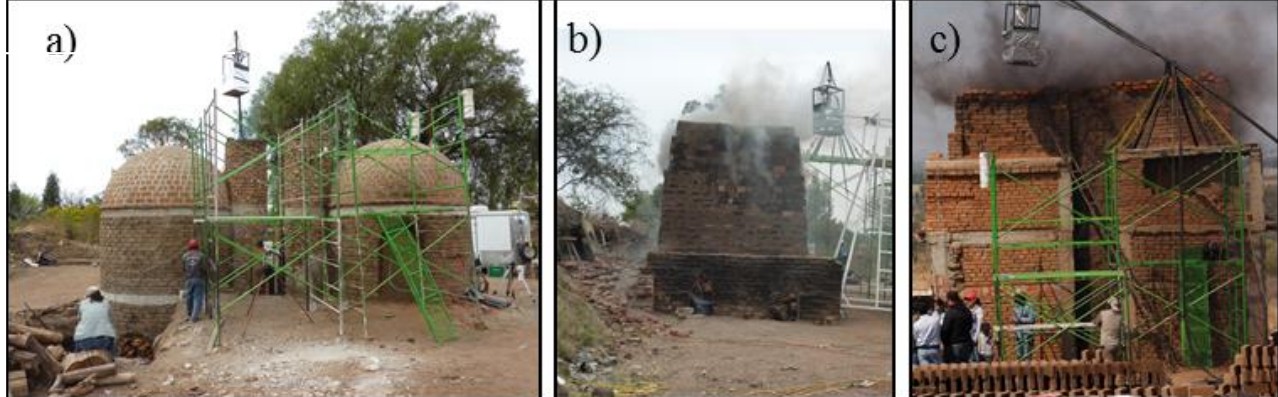

**Figure 1. Brick kilns sampled: a) MK2 kiln in El Refugio, Guanajuato, b) Traditional-campaign kiln in El Refugio, Guanajuato, c) Traditional-fixed kiln in Abasolo, Guanajuato.**





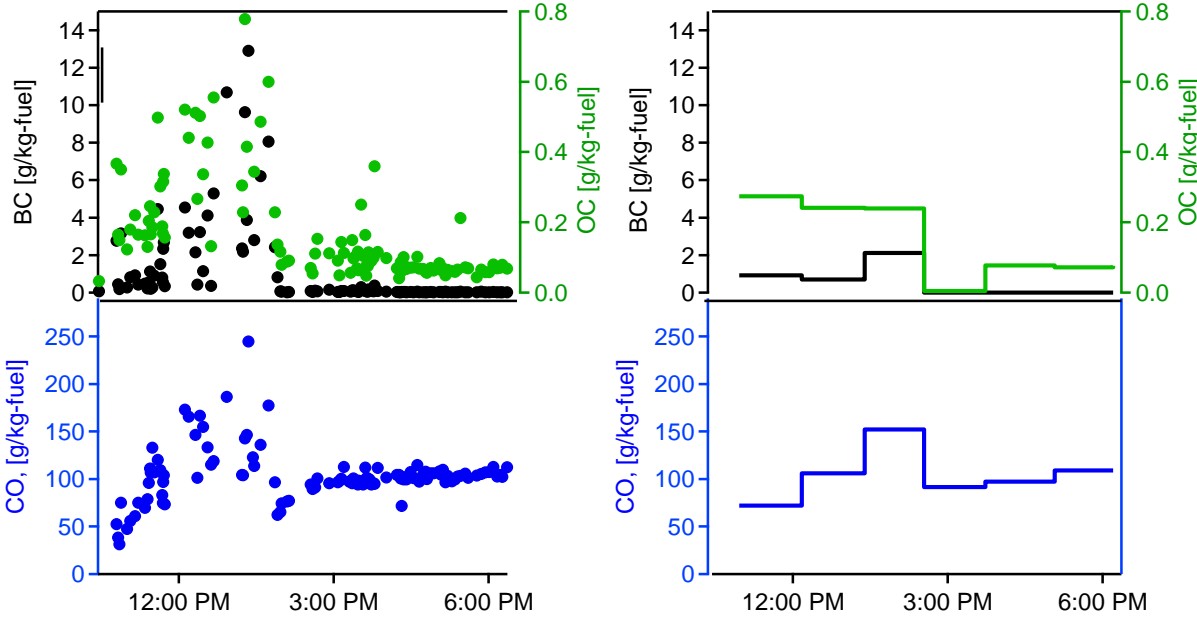

**Figure 2. Temporal profiles of fuel-based CO, BC, and OC emission factors (g/kg-fuel) for the traditional-fixed kiln obtained with the AML and the tracer ratio technique (left panels) and with the sampling probe technique (right panels).**



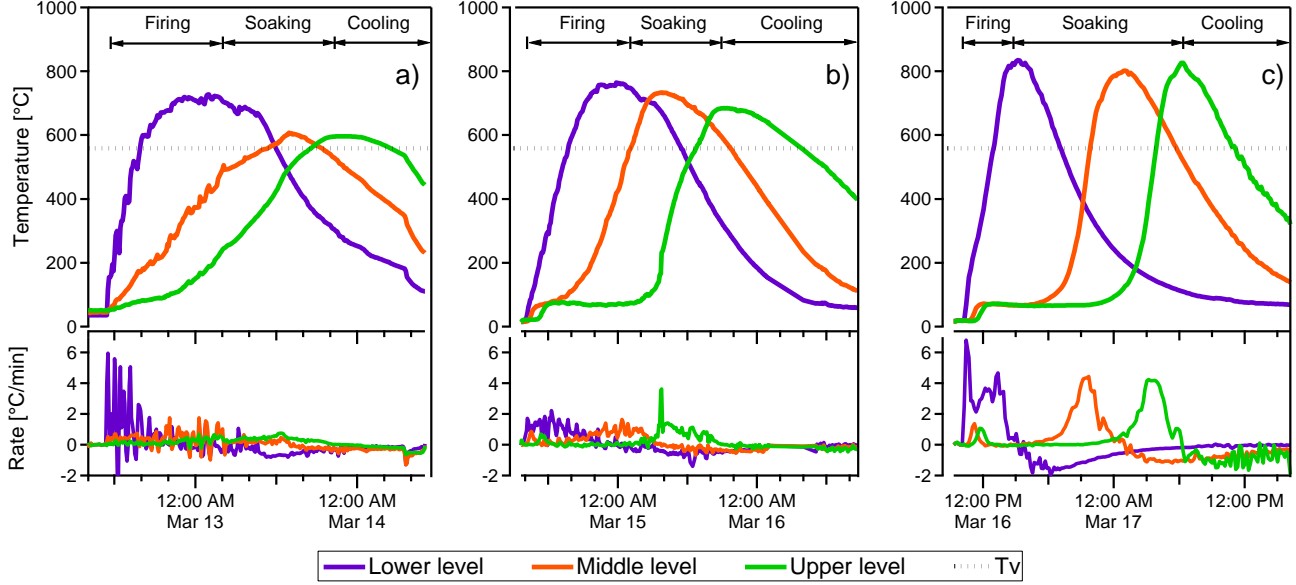

**Figure 3: Temperature profiles (top panels) and temperature change rates (bottom panels) for: a) MK2, b) Traditional-campaign, and c) Traditional-fixed brick kilns for the lower, middle, and upper levels of the kilns. Horizontal dotted line represents $T_v$, the**
5 **temperature for the quartz inversion process (573 ºC). The figures also indicate the stages of burning, smoldering, and cooling for each kiln, as defined in the text.**





**Figure 4: Inter-comparison of emission factors normalized to the average of the three kilns by pollutant for a) CO, $CO_2$, NO, $NO_2$, OC, BC, $PM_{2.5}$, $CH_4$; b) sampled VOC species; and c) inorganic components. Panel d) compares the modified combustion efficiency, specific energy consumption, fuel consumption and brick quality for the three sampled kilns.**