# Peer review of "Black Carbon, Organic Carbon, and Co-pollutants Emissions and Energy Efficiency from Artisanal Brick Production in Mexico"

_Atmospheric Chemistry and Physics, 2017_

## Referee Comment (RC1) · C. Bruce (Referee) · 15 Jan 2018

Comments on manuscript: acp-2017-1154, Atmospheric Chemistry and Physics Worldwide communications/publications on the environmental properties of brick kiln burns are progressively more detailed (in the sense of number of environmental pollutants measured) and analytic, this paper follows this pattern. Nevertheless, the authors mention a problem of which this reviewer is quite aware and which represents the primary weak point of this paper: variability between burns for a number of reasons, mentioned too briefly in this paper. This variability has been noted in other papers of the author's recent reviews, in kiln research from South Africa and Vietnam, for exam-

ple. For this paper the problem is more severe than for the others since only one burn for each of several variations of kilns was performed. Either the procedures / materials / specific construction must be specified or. . .a number of complete burns must be monitored to be definitive about various features and quantities. . .anything less is not definitive of the characteristics. What can be done? The authors can make the point more clearly that this represents a sampling and is not a definitive comparative description: to compare, fuel/stacking/similar clays/ brick additives/feeding procedures or knowledge of aging of the kilns (# of previous burns in the same kiln) were not standardized nor described. Complete burns On other issues: 1. Sampling downstream: many papers have made attempts to quantify dispersion in two dimensions. I have to think that they do not understand dispersion theory, another very poorly defined result. Could have been done much better. Next time profile the downwind cross section using instrumented drones or other methods to be more definitive. Just not worthwhile as performed here. 2. What supplemental documents? This article should stand on its own or on previous publications. 3. Data on the MK2 will depend on development in three defined intervals; pre-switch, transition to coupled kilns and final the coupled burn. . .all very distinct and not even discussed. Frequency of sampling? Quality of temporal integration? 4. Chemistry: nicely done and informative. . . Just with the previous concerns for representation. 5. Temperature profiles were puzzling, too sparse to analyze. 6. Spatial representation of the outflow requires time to average in each location. Uncertain how well performed. Like many points is undefined. In summary, should be presented as a first try at comparison but not as definitive.

---

## Referee Comment (RC2) · Anonymous Referee #2 · 14 Mar 2018

It is an interesting and well prepared field monitoring work for obtaining the emission factors of various pollutants and PM associated chemical components from brick kilns. It is very helpful for the emission inventory updating and related human health risk assessment research. Also considering the difficulty and the complexity of this type of motoring works, I highly recommend the publication of this manuscript after the following questions are answered and corresponding revisions are done. (1) Please give a sampling frame figure, after Figure 1, to clear show the size of the brick kilns, the relative location of the sampling probe, the relative location of the AML. It is very important for the comparison of the emission factors as the location reflects the dilution extent of the flumes, considering quantitative dilution effect could not be obtained in

this study. (2) The range of 20-100 m is huge enough for the chemical evolution and variation of flumes. How the authors consider its impact on the emission factors? In Table 1, the OC emission factors are quite different for the two methods. The authors should give clear suggestions that when establishing the emission inventory, which emission factors should be selected. (3) I am not clear about how the author obtain the release flow rate of tracers. By AML, you can just obtain the emission concentrations, but not the flow rate information. (4) I wonder whether the emission concentrations is too high for the detection of SP-AMS. Please give detailed operating procedures for the switch of BC and other components monitoring by SP-AMS during the whole sampling period. (5) Whether the MCE is of significant differences between the MK2 and Traditional kilns. The one cycle test in this study may be limited. The authors should better describe this.

---

## Author Comment (AC1) · 6 Apr 2018

**Letter to the Editor:**

Dear Editor,

We would like to thank the two reviewers for their careful reading and constructive comments.

We are attaching our final response to the reviewers' comments and the revised manuscript, which incorporates the comments and changes suggested by the reviewers.

Thank you in advance for your consideration.

Best regards,

Luisa Molina

**Responses to Reviewer 1**

Worldwide communications/publications on the environmental properties of brick kiln burns are progressively more detailed (in the sense of number of environmental pollutants measured) and analytic, this paper follows this pattern. Nevertheless, the authors mention a problem of which this reviewer is quite aware and which represents the primary weak point of this paper: variability between burns for a number of reasons, mentioned too briefly in this paper. This variability has been noted in other papers of the author's recent reviews, in kiln research from South Africa and Vietnam, for example. For this paper the problem is more severe than for the others since only one burn for each of several variations of kilns was performed. Either the procedures / materials / specific construction must be specified or. . .a number of complete burns must be monitored to be definitive about various features and quantities. . .anything less is not definitive of the characteristics. What can be done? The authors can make the point more clearly that this represents a sampling and is not a definitive comparative description: to compare, fuel/stacking/similar clays/ brick additives/feeding procedures or knowledge of aging of the kilns (# of previous burns in the same kiln) were not standardized nor described. Complete burns

**Response**

We thank the reviewer, Dr. Charles Bruce, for his constructive comments on this paper.

The main concern expressed by the reviewer is the inherently large variability of the brick production process. As the reviewer pointed out, a small sampling size is a common issue in the scarcely available literature on brick kilns emissions characterizations worldwide. This is perhaps in part due to the considerable logistical complexity of real world measurements for these sources, but also to the large combination of materials, fuels, kiln types, and operational practices that brick producers use. This further highlights the strong need to increase the number of databases on locally-measured emission characteristics of brick kilns.

As the reviewer pointed out, this issue can be addressed in two ways: either by describing the particular parameters (materials, fuels, operational practices) of the brick production process for each kiln sampled, or by attempting to standardize the processes for the purpose of measurements. However, artisanal production of bricks is by definition not standardized as it depends on the variations of kiln types and burning practices that are generationally learned by producers, and adjusted by clay type and fuel availability. Therefore, we believe it is better to describe in more detail the conditions of the brick-making processes during samplings, as these represent artisanal combustion processes. This will also facilitate future comparisons with other results.

As suggested, we have now included more detailed descriptions of the clay, fuels, and additives used as well as fuel feeding practices during the sampling of the brick kilns. The descriptions have been added in section 6 of the Supplement Material document to keep the readability in the main manuscript. We have also expanded the discussions on the reasons for the variability between burns and further

clarified that the results are not intended to be definitive generalizations of the brick making process but to help in the understanding of the effects of different kiln designs and fuels on gaseous and particulate phase emissions from brick kilns.

As an additional note, although due to the nature of the brick-making process it is not possible to standardize the burning practices for sampling purposes, it is certainly possible to standardize the sampling techniques and analysis protocols used. We recommend this practice in our study by using the methodology recently proposed by the Climate and Clean Air Coalition Brick Production Initiative. This will allow further reducing the uncertainty during the comparison of our results with future studies.

On other issues: 1. Sampling downstream: many papers have made attempts to quantify dispersion in two dimensions. I have to think that they do not understand dispersion theory, another very poorly defined result. Could have been done much better. Next time profile the downwind cross section using instrumented drones or other methods to be more definitive. Just not worthwhile as performed here.

**Response**

The reviewer's comment suggests that some clarification may be needed in the Methods section. In this paper we have not attempted to quantify the dispersion of air pollutants or obtain the cross section or any sort of spatial representation of the brick kilns emission plumes. We have further clarified this by explicitly stating in the methods section that the well-established plume tracer ratio technique is not meant to address the aforementioned issues but to quantify the emission rates of co-emitted pollutants from a single source. Although the technique has been successfully applied multiple times to other sources (e.g., wastewater treatment plants, industrial stacks, etc.), to our knowledge this technique had never been applied for measuring emissions from brick production and therefore it is worthwhile to compare it with the more traditional filter-based technique.

2. What supplemental documents? This article should stand on its own or on previous publications.

**Response**

We believe the additional materials provided in the supplement material document will be valuable to the readers. The supplemental material document includes: 1) descriptions of the fuel types and analysis of chemical compositions of fuels and materials used for each kiln; 2) technical characteristics of instrumentation used; 3) additional results on the mechanical resistance analysis of the bricks and emission factors obtained. We have further added a detailed description of the artisanal brick-making process for each kiln. We believe this additional information will allow the reader to more completely understand the context of the samplings.

3. Data on the MK2 will depend on development in three defined intervals; pre-switch, transition to coupled kilns and final the coupled burn. . .all very distinct and not even discussed. Frequency of sampling? Quality of temporal integration?

**Response**

As described above, detailed information on the artisanal brick-making procedures for the sampled kilns, including the MK2 kiln, has now been added in the Supplemental Material document. The sampling frequency and temporal integration of the samplings are already described in the Methods section.

4. Chemistry: nicely done and informative. . . Just with the previous concerns for representation.

**Response**

We appreciate the comment from the reviewer. As stated above and in the manuscript, the results of this study should not be considered as generalizations of brick production practices due to the various combination of materials, fuels, kiln types, and operational practices that brick producers use. However, we believe that the results contribute to the understanding of the chemical characterization of emissions from brick kilns and represent valuable additions to the currently scarce literature.

**5. Temperature profiles were puzzling, too sparse to analyze.**

**Response**

As mentioned in the manuscript, temperature profiles were obtained at the lower, middle, and upper levels of each kiln using thermocouples and the average results are presented in Figure 3. We have now further clarified in the methods and results sections that the temperature measurements were obtained using four thermocouples at each of the three levels. In other words, we obtained three cross-sections (each at a different level) using 12 temperature measurements for each kiln. The analyses and discussions on the average temperature profiles that we present in the manuscript are directed towards: 1) defining the stages of the brick production process for our chemical analysis purposes, 2) understanding the relations between vertical changes in heating rates inside the kiln and the mechanical resistance of the produced bricks.

As a corollary, our results indicate that the spatial distribution of the internal temperature in the kiln is closely related to quality of the products. Thus we suggest that improving the structural design and thermal energy transfer of brick kilns could be an alternative way of increasing the efficiency of brick kiln production.

6. Spatial representation of the outflow requires time to average in each location. Uncertain how well performed. Like many points is undefined. In summary, should be presented as a first try at comparison but not as definitive.

**Response**

Our study is not directed towards obtaining the spatial representation of the brick kiln emissions outflow and this has now been clarified in the methods section.

"The second technique used to sample the kilns was based on the tracer ratio method in which the emission rate of the targeted source is obtained by simultaneously measuring in real-time the abovebackground concentrations of the species of interest and of a selected gas tracer with a known release rate that is co-located at the emission source (Lamb et al., 1995). This method is based on the fundamental assumption that a relatively unreactive mixture of gases emitted from a common location experiences a quasi-perfect co-dispersion and equivalent dilution through the atmosphere. The tracer ratio method does not quantify the dispersion of air pollutants or the spatial representation of the brick kilns emission plumes, but is used to quantify the emission rates of co-emitted pollutants from a single source."

**Responses to Reviewer 2**

It is an interesting and well prepared field monitoring work for obtaining the emission factors of various pollutants and PM associated chemical components from brick kilns. It is very helpful for the emission inventory updating and related human health risk assessment research. Also considering the difficulty and the complexity of this type of motoring works, I highly recommend the publication of this manuscript after the following questions are answered and corresponding revisions are done.

**Response**

We thank the reviewer for the thoughtful comments on the paper.

(1) Please give a sampling frame figure, after Figure 1, to clear show the size of the brick kilns, the relative location of the sampling probe, the relative location of the AML. It is very important for the comparison of the emission factors as the location reflects the dilution extent of the flumes, considering quantitative dilution effect could not be obtained in this study.

**Response**

We have included in the Supplemental Material document additional figures showing the locations of the samplings and the spatial location of the AML with respect to the kilns. These figures were included in the Supplemental Material document to keep the readability in the manuscript. We have also included additional information on the artisanal brick making procedures for each sampled kiln.

(2) The range of 20-100 m is huge enough for the chemical evolution and variation of flumes. How the authors consider its impact on the emission factors? In Table 1, the OC emission factors are quite different for the two methods. The authors should give clear suggestions that when establishing the emission inventory, which emission factors should be selected.

**Response**

The typical distances of the AML to the kilns ranged from about 20-100 m, depending on the feasibility to "find" the plume. As stated in the manuscript, the calculation of the emission factors using the tracer sampling technique does not requires the estimation of the dilution of the plumes because the analysis is based on the ratios of the targeted species and the tracer. This is possible due to the very low detection and high precision limits of the instruments on-board the AML. However, as pointed out by the reviewer it is possible that some condensation of SVOCs occurs as the plume cools, thereby increasing the OC content into the particle phase. One possibility to quantify this effect for a future experiment would be through mass balance while measuring individual VOCs both directly at the emission point and downwind together with OC, but this is beyond the scope of our paper. We have now included a paragraph acknowledging this effect and a note in the results tables (Tables 2 and 3). The following paragraph has been added for clarification:

"In this study, condensation of emitted semi-volatile VOCs between the top of the kiln and the sampling location of the mobile laboratory downwind the plume is possible due to the strong temperature gradient, adding organic content to the measured OC. However, quantification of this effect is beyond the scope of this study."

"Results for OC and VOCs obtained with the tracer ratio method include the effects of possible condensation of organics into the particle phase."

(3) I am not clear about how the author obtain the release flow rate of tracers. By AML, you can just obtain the emission concentrations, but not the flow rate information.

**Response**

We thank the reviewer for pointing out the need to clarify the measurement of the release flow of the tracers. The following paragraph has been added in the manuscript:

"The tracer gas was released from a compressed gas cylinder of pure N2O located in a separate vehicle. The flow rate was controlled with an MKS mass flow controller (MFC), which was calibrated against a traceable Drycal mass flowmeter several times over the course of the measurement campaign. A 3/8" polyethylene tube extended from the mass flow controller to the desired location, allowing the cylinder and MFC to be located in a close but safe distance from the kiln. Mass flow rates were digitally recorded, and manually logged."

(4) I wonder whether the emission concentration is too high for the detection of SP-AMS. Please give detailed operating procedures for the switch of BC and other components monitoring by SP-AMS during the whole sampling period.

**Response**

The SP-AMS is able to handle high concentrations of particulate matter routinely. It is frequently used in source studies where organics and black carbon may be higher than 100  $\mu$ g/m3 (Massoli et al., 2012; Zavala et al., 2017). The SP-AMS does use the same laser as an SP2 system but it uses the laser for a different purpose. In the case of the SP-AMS it is simply using the laser to heat the particle beam creating gas phase molecules which are then ionized by reaction with electrons being emitted by a tungsten filament. This measurement technique is really not negatively impacted as concentrations increase.

Regarding switching related to BC there is no switch between measurement modes. The following paragraph has been added in the Methods section:

"The SP-AMS acquires data in 1 second mode during which it obtains an average mass spectrum sampling from 12-1000 m/z. The mass spectrum is then processed and high resolution fits are applied to

peaks (e.g., C3 at m/z 36 or C3H7 at m/z 43) to distinguish between BC and organics. All fit peaks are summed, counted as a particular species and then that species is quantified for each second. There are 2 vaporizers simultaneously heating particles so that gas phase molecules are then available to ionize by reaction with electrons. The laser vaporizer heated particles with a 1064-nm laser while the conventional AMS vaporizer was also present and after passing through the laser vaporizer the particle beam impacted the conventional vaporizer which was heated to 600 °C. Inorganic species such as sulfate and nitrate are not vaporized by the 1064-nm laser but were vaporized by a conventional heater."

1Molina Center for Energy and the Environment, La Jolla, CA, 92037, USA 2GAMATEK, Monterrey, Nuevo Leon, Mexico 3Desert Research Institute, Las Vegas, NV, 89119, USA 4Universidad Autónoma Metropolitana, Mexico City, Mexico

Correspondence to: Luisa Molina (ltmolina@mce2.org; ltmolina@mit.edu)

- 15 Abstract. In many parts of the developing world and economies in transition, small-scale traditional brick kilns are a notorious source of urban air pollution. Many are both energy inefficient and burn highly polluting fuels that emit significant levels of black carbon (BC), organic carbon (OC) and other atmospheric pollutants into local communities, resulting in severe health and environmental impacts. However, only a very limited number of studies are available on the emission characteristics of brick kilns; thus there is a need to characterize their gaseous and particulate matter (PM) emission factors to better assess their
- 20 overall contribution to emissions inventories and to quantify their ecological, human health, and climate impacts. In this study, the fuel-, energy-, and brick-based emissions factors and time-based emission ratios of BC, OC, inorganic PM components, CO, SO2, CH4, NOx, and selected volatile organic compounds (VOCs) from three artisanal brick kilns with different designs in Mexico were quantified using the tracer ratio sampling technique. Simultaneous measurements of PM components, CO and  $CO_2$  were also obtained using a sampling probe technique. Additional measurements included the internal temperature of the
- 25 brick kilns, mechanical resistance of bricks produced, and characteristics of fuels employed. Average fuel-based BC emission factors ranged from 0.15 - 0.58 g/kg-fuel whereas BC/OC mass ratios ranged from 0.9 - 5.2, depending on the kiln type. The results show that both techniques capture similar temporal profiles of the brick kiln emissions and produce comparable emission factors. A more integrated inter-comparison of the brick kilns' performances was obtained by simultaneously assessing emissions factors, energy efficiency, fuel consumption, and the quality of the bricks produced.

**1** Introduction 30**

Artisanal clay brick production using small-scale traditional kilns is a highly polluting activity occurring in developing countries and economies in transition to manufacture building materials. Moreover, traditional brick production is a serious local health hazard to the residents of the poor neighborhoods that typically host brickyards, as well as to brick makers

5

<sup>5Secretaria del Medio Ambiente, Mexico City, Mexico 10 6Aerodyne Research, Inc., Billerica, MA, 01821, USA 7Department 
[revised manuscript text omitted]

|                                | MK2     |         | Trac  | litional-camp | aign    | Tı     | raditional-fix | ed      |        |
|--------------------------------|---------|---------|-------|---------------|---------|--------|----------------|---------|--------|
|                                | SP      | AN      | 1L    | SP            | AN      | /IL    | SP             | AN      | /IL    |
|                                | EF-fuel | EF-fuel | ER    | EF-fuel       | EF-fuel | ER     | EF-fuel        | EF-fuel | ER     |
| MCE                            | 0.96    | 0.94    |       | 0.95          | 0.94    |        | 0.91           | 0.92    |        |
| MCE                            | (0.02)  | (0.04)  |       | (0.02)        | (0.03)  |        | (0.02)         | (0.02)  |        |
| CO                             | 1583    | 1595    |       | 1527          | 1597    |        | 1668           | 1658    |        |
| CO                             | (28)    | (58)    |       | (28)          | (54)    |        | (40)           | (43)    |        |
| CO                             | 44.4    | 65.4    | 270.7 | 50.5          | 65.3    | 553.7  | 105.2          | 105.3   | 8500.2 |
| $TOC^2$                        | (18)    | (526)   | (902) | (17)          | (43)    | (1040) | (24)           | (36)    | (9588) |
| $TOC^2$                        | 2.0     |         |       | 5.0           |         |        | 14.6           |         |        |
| 1003                           | (2)     |         |       | (4)           |         |        | (2)            |         |        |
| CU                             |         | 2.39    | 11.90 |               | 3.34    | 28.0   |                | 5.92    | 551    |
| CH4                            |         | (2.6)   | (28)  |               | (2.9)   | (53)   |                | (2.2)   | (699)  |
| NO                             |         | 1.02    | 4.3   |               | 1.05    | 13.8   |                | 0.76    | 43.4   |
| NO                             |         | (0.9)   | (8)   |               | (2.1)   | (71)   |                | (0.3)   | (31)   |
| NO                             |         | 1.7     | 7.4   |               | 0.93    | 7.8    |                | 1.01    | 53.8   |
| NO 2                |         | (1.8)   | (18)  |               | (1.4)   | (27)   |                | (0.6)   | (36)   |
| SO 2                |         | 1.0     | 3.6   |               | 0.27    | 1.1    |                | 0.13    | 8.7    |
|                                |         | (1.4)   | (8)   |               | (0.3)   | (2)    |                | (0.1)   | (9)    |
| PM 2.5 3 | 1.94    | 1.66    | 3.9   | 4.62          | 2.28    | 17.5   | 1.32           | 1.26    | 171.9  |
|                                | (0.6)   | (0.8)   | (2)   | (4.3)         | (1.8)   | (15)   | (1.3)          | (2.2)   | (152)  |
| 2.0                            | 0.15    | 0.67    | 1.6   | 0.28          | 0.73    | 3.4    | 0.54           | 1.03    | 149.4  |
| BC                             | (0.1)   | (0.5)   | (3)   | (0.2)         | (0.6)   | (5)    | (0.8)          | (2.2)   | (377)  |
| 1                              | 0.03    | 0.52    | 1.5   | 0.3           | 1.18    | 12.6   | 0.14           | 0.18    | 19.5   |
| $OC^{\underline{4}}$           | (0.03)  | (0.6)   | (5)   | (0.7)         | (1.7)   | (38)   | (0.1)          | (0.2)   | (31)   |
|                                |         | 31.0    | 84.5  |               | 26.8    | 150.1  |                | 8.1     | 912.4  |
| Fullerene (x10 -3 ) |         | (39)    | (183) |               | (30)    | (252)  |                | (12)    | (1820) |
|                                | 246.3   | 96.8    | 155.2 | 942.2         | 66.7    | 240.6  | 2.9            | 6.6     | 312.5  |
| Ammonium (x10 -3 )  | (105)   | (64)    | (225) | (1068)        | (69)    | (442)  | (2)            | (5)     | (349)  |
|                                | 1.4     | 23.4    | 68.4  | 10.7          | 12.3    | 69.7   | 3.4            | 4.8     | 393.6  |
| Nitrate (x10 -3 )   | (1)     | (33)    | (173) | (17)          | (20)    | (197)  | (2)            | (6)     | (691)  |
|                                | 135.5   | 121.3   | 315.1 | 91.9          | 68.8    | 302.1  | 48.8           | 21.6    | 1721.7 |
| Sulfate (x10 -3 )   | (96)    | (203)   | (678) | (36)          | (104)   | (620)  | (59)           | (19)    | (2240) |
|                                | 617.2   | 234.1   | 305.2 | 1956.3        | 226.8   | 936.8  | 21.7           | 10.5    | 649.7  |
| Chloride $(x10^{-3})$          | (200)   | (153)   | (397) | (2095)        | (227)   | (1766) | (14)           | (4)     | (624)  |
|                                | 27.2    | ( )     |       | 21.9          |         | (,     | 15.3           | ( )     |        |
| Sodium (x10 -3 )    | (15)    |         |       | (9)           |         |        | (12)           |         |        |
|                                | 0.67    |         |       | 1.34          |         |        | 1.11           |         |        |
| Magnesium $(x10^{-3})$         | (0.4)   |         |       | (0.8)         |         |        | (0.8)          |         |        |
| -                              | 189.3   |         |       | 114.7         |         |        | 44.0           |         |        |
| Potassium (x10 -3 ) | (146)   |         |       | (74)          |         |        | (65)           |         |        |
| Calcium (x10 -3 )   | 5.5     |         |       | 7.9           |         |        | 5.3            |         |        |

Table 2. Average modified combustion efficiency (MCE), fuel-based emission factors EF [g/kg fuel], and emission rates ER [g/min] obtained with the sampling probe (SP) and tracer ratio (AML) techniques.1

[revised manuscript text omitted]

|                                               |             | This study?              |                   | Stockwell et al., (2016), |                       | Christian et al. (2010) Mavias   |
|-----------------------------------------------|-------------|--------------------------|-------------------|---------------------------|-----------------------|----------------------------------|
|                                               |             | This study               |                   | Jayarathne e              | et al., (2017), Nepal | Christian et al., (2010), Mexico |
| kiln type                                     | MK2         | Traditional-
campaign | Traditional-fixed | Clamp                     | Forced draft zig Zag  | Traditional-fixed                |
| Fuels                                         | Wood        | Wood                     | Wood, sawdust     | Coal,
hardwood         | Coal, bagasse         | Sawdust, wood waste              |
| PM 2.5                             | 1.94 (0.6)  | 4.62 (4.3)               | 1.32 (1.3)        | 10.7 (1.6)                | 15.1 (3.7)            | $1.2-2.0^{ m bb}$                |
| BC                                            | 0.15 (0.1)  | 0.28 (0.2)               | 0.54 (0.8)        | 0.0172                    | 0.112                 | 0.596-1.5                        |
| OC                                            | 0.03 (0.03) | 0.3 (0.7)                | 0.14 (0.1)        | 6.74                      | 1.05                  | 0.073-0.283                      |
| $SO_2$                                        | 1. (1.4)    | 0.27 (0.3)               | 0.13 (0.1)        | 13                        | 12.7                  |                                  |
| СО                                            | 44.4 (17.7) | 50.5 (16.7)              | 105.2 (24.3)      | 70.9                      | 10.1                  | 25.7-55.7                        |
| CO 2                               | 1582 (28)   | 1526 (28)                | 1668 (40)         | 2102                      | 2620                  | 1736-1787                        |
| NO                                            | 1.02 (0.9)  | 1.05 (2.1)               | 0.76 (0.3)        | bdl                       | 1.28                  |                                  |
| NO 2                               | 1.7 (1.8)   | 0.93 (1.4)               | 1.01 (0.6)        | 0.297                     | 0.0821                |                                  |
| $CH_4$                                        | 2.39 (2.6)  | 3.34 (2.9)               | 5.92 (2.2)        | 19.5                      | 0.0873                | 1.13-2.16                        |
| $C_2H_6$                                      | 0.15 (0.2)  | 0.21 (0.2)               | 0.44 (0.1)        | 5.37                      | 0.00206               |                                  |
| CH 3 OH                            | 1.99 (2.)   | 1.19 (2.3)               | 3.25 (1.2)        | 1.77                      | 0.112                 | 0.39-1.42                        |
| CH 3 COOH                          | 2.64 (3.1)  | 0.89 (2.6)               | 1.04 (0.8)        | 0.43                      | 0.471                 | 0.21                             |
| $C_6H_6$                                      | 0.84 (0.9)  | 0.66 (0.7)               | 0.5 (0.3)         | 1.68                      | 0.00825               |                                  |
| C 6 H 5 CH 3 | 0.93 (0.8)  | 0.42 (0.9)               | 0.28 (0.2)        | 1.05                      | 0.0028                |                                  |
| C 3 H 6 O               | 1.28 (1.5)  | 0.61 (1.9)               | 0.91 (0.3)        | -                         | 0.146                 |                                  |
| $C_2H_4O$                                     | 1.13 (1.2)  | 0.54 (0.4)               | 2.18 (0.5)        | 0.0413                    | 0.0694                |                                  |

Table 4. Comparison of fuel-based emission factors (g/kg-fuel) measured in this study with other studies.

a Numbers in parenthesis represent 1 standard variation. Values in Crhistian et al, (2010) represent range of averages. "bdl" indicates below detection limit; "-" indicates concentrations were not greater than background.

5 <sup>b Estimated from measurements of OC, EC, metals, and ions (but not sulfate).